# Proton conductance by human uncoupling protein 1 is inhibited by purine and pyrimidine nucleotides

Scott A Jones [ID][1], Alice P Sowton [ID][1], Denis Lacabanne [ID][1], Martin S King [ID][1], Shane M Palmer [ID][1], Thomas Zögg [ID][2,3], Els Pardon[2,3], Jan Steyaert [ID][2,3], Jonathan J Ruprecht [ID][1] & Edmund R S Kunji [ID][1✉]

## Abstract

**Uncoupling protein 1 (UCP1, SLC25A7) is responsible for the thermogenic properties of brown adipose tissue. Upon fatty acid activation, UCP1 facilitates proton leakage, dissipating the mitochondrial proton motive force to release energy as heat. Purine nucleotides are considered to be the only inhibitors of UCP1 activity, binding to its central cavity to lock UCP1 in a proton-impermeable conformation. Here we show that pyrimidine nucleotides can also bind and inhibit its proton-conducting activity. All nucleotides bound in a pH-dependent manner, with the highest binding affinity observed for ATP, followed by dTTP, UTP, GTP and CTP. We also determined the structural basis of UTP binding to UCP1, showing that binding of purine and pyrimidine nucleotides follows the same molecular principles. We find that the closely related mitochondrial dicarboxylate carrier (SLC25A10) and oxoglutarate carrier (SLC25A11) have many cavity residues in common, but do not bind nucleotides. Thus, while UCP1 has evolved from dicarboxylate carriers, no selection for nucleobase specificity has occurred, highlighting the importance of the pH-dependent nucleotide binding mechanism mediated via the phosphate moieties.**

**Keywords** Uncoupling Protein; Thermogenesis; Bioenergetics; SLC25; Pyrimidine Nucleotides
**Subject Categories** Metabolism; Structural Biology

## Introduction

Mammalian brown adipose tissue (BAT) is central to classical non-shivering thermogenesis (Cannon and Nedergaard, 2004). The ability of BAT to maintain core body temperature against cold exposure is due to the highly expressed uncoupling protein 1 (UCP1) in its mitochondria. When activated by free fatty acids, UCP1 facilitates the movement of protons from the intermembrane space into the mitochondrial matrix, bypassing ATP synthase, to dissipate the proton motive force. This activity results in the

metabolic energy produced by the oxidation of sugars and fats being converted to heat (Lidell, 2019; Harms and Seale, 2013). For this reason, UCP1 is considered to be a target for the treatment of obesity and related metabolic illnesses (Chondronikola et al, 2014; Nedergaard et al, 2007; Saito et al, 2009; Cypess et al, 2009).

UCP1 is a member of the SLC25 mitochondrial carrier family, which largely consists of solute carriers involved in metabolite transport across the mitochondrial inner membrane (Ruprecht and Kunji, 2020). The family members share a common structural fold consisting of six transmembrane and three matrix helices (Pebay-Peyroula et al, 2003; Ruprecht et al, 2014, 2019). Mitochondrial carriers have an alternating access mechanism (Cimadamore-Werthein et al, 2023, 2024) and cycle between two defined states; one in which the central cavity is open to the cytoplasm (cytoplasmic-open or c-state) (Ruprecht et al, 2014; Pebay-Peyroula et al, 2003) and another where it is open to the mitochondrial matrix (matrix-open or m-state) (Ruprecht et al, 2019). The opening and closing of the central cavity is determined by disruption and formation of two interaction networks. Formation of the matrix salt bridge network and glutamine braces closes the matrix side in the c-state, whereas formation of the cytoplasmic salt bridge network and tyrosine braces closes the cytoplasmic side in the m-state (Ruprecht et al, 2019; King et al, 2016; Ruprecht and Kunji, 2021). State interconversion is triggered by substrate binding to contact point residues on even-numbered transmembrane helices within the central cavity (Kunji and Robinson, 2006; Robinson and Kunji, 2006; Robinson et al, 2008). UCP1 contains all the functional elements of SLC25 family members and, therefore, could have the same structural mechanism (Crichton et al, 2017; Jones et al, 2024).

It is widely accepted that UCP1 activity is regulated by cytosolic purine nucleotides (Rial et al, 1983; Lin and Klingenberg, 1982; Heaton and Nicholls, 1977; Heaton et al, 1978). As purine nucleotides are present in high concentrations in the cytoplasm (Traut, 1994), UCP1 activation only occurs in the presence of molecules such as free fatty acids. The structural basis of UCP1 inhibition has recently been advanced by the electron cryo-microscopy (cryo-EM) structures of human UCP1 bound to guanosine triphosphate (GTP) (Jones et al, 2023) and adenosine triphosphate (ATP) (Kang and Chen, 2023). Both purine nucleotides bind to UCP1 via similar interactions within the positively charged central cavity, generating a c-state with a large insulation

[1]MRC Mitochondrial Biology Unit, University of Cambridge, Cambridge Biomedical Campus, Keith Peters Building, Cambridge CB2 0XY, UK. [2]VIB-VUB Center for Structural Biology, VIB, Pleinlaan 2, B-1050 Brussels, Belgium. [3]Structural Biology Brussels, Vrije Universiteit Brussel, Pleinlaan 2, B-1050 Brussels, Belgium. ✉E-mail: ersk2@cam.ac.uk

layer to prevent proton conductance (Jones et al, 2023; Kang and Chen, 2023). The nucleotide-binding site consists of ionic interactions with the positively charged matrix network residues and three arginine residues at the substrate contact points (Jones et al, 2023; Kang and Chen, 2023). In addition, both purine ring systems form a cation-π interaction with R92. The structures also reveal how purine nucleotides act as inhibitors; the nucleotide phosphate moiety (β and γ) interacts directly with the matrix gate of UCP1 preventing the matrix network from breaking, blocking state interconversion. Thus, purine nucleotide binding creates a proton-impermeable layer between the water-accessible central cavity and the mitochondrial matrix (Jones et al, 2024, 2023). Interactions between these nucleotides and UCP1 differ at the nucleobase: guanine binds via electrostatic bonds with E191 and hydrogen bonds with N188 (Jones et al, 2023), while adenine binds via hydrogen bonds with N188 and N282 (Kang and Chen, 2023; Jones et al, 2024). These purine nucleotides interact with the contact point residues, matrix gate residues, and residues towards the cytoplasmic side of the cavity, differing from substrate binding to SLC25 family members, which interacts only with residues around the contact points to trigger conformational changes (Jones et al, 2024; Mavridou et al, 2022). The interaction of purine nucleotides with UCP1 is also known to be pH-dependent (Echtay et al, 2018; Klingenberg, 1988; Jones et al, 2023; Lee et al, 2015). The resolved nucleotide-bound UCP1 structures show that the negative charges of the matrix network are within bonding distance of the negatively charged phosphate moiety (β and γ). They can form interactions, but only when protons are involved, providing an explanation for the pH-dependent binding mechanism (Jones et al, 2024, 2023).

The inhibition of UCP1 is widely believed to be specific to purine nucleotides (Heaton and Nicholls, 1977; Divakaruni et al, 2012; Gagelin et al, 2023). These observations were based on competition assays between pyrimidine nucleotides and radiolabelled guanine nucleotides (Heaton and Nicholls, 1977) or fluorescent guanine nucleotides in whole mitochondria (Divakaruni et al, 2012). In both cases, cytosine nucleotides (CTP or CDP) were unable to displace the binding of GDP (Heaton and Nicholls, 1977; Divakaruni et al, 2012), while uridine triphosphate (UTP) could only weakly compete with GDP binding (Heaton and Nicholls, 1977). In addition, UTP inhibition of proton conductance was significantly lower than that of purine nucleotides (Heaton and Nicholls, 1977). These studies indicated that pyrimidine nucleotides bind to UCP1 with much lower affinity than purine nucleotides, limiting their relevance to UCP1 regulation under physiological conditions. Selectivity of purine nucleotides was recently analysed using molecular dynamic simulations, which compared the binding of UDP and GDP in an attempt to explain the nucleotide specificity of UCP1 inhibition (Gagelin et al, 2023). Although the poses are significantly different from those in the determined structures (Jones et al, 2023; Kang and Chen, 2023), they do show similar contacts between the phosphate and ribose groups of GDP and UDP (Gagelin et al, 2023), suggesting that pyrimidine nucleotides might be able to bind. Moreover, both ATP and GTP bind with high affinity despite having different interactions at the nucleobase (Jones et al, 2023; Kang and Chen, 2023).

Here, we investigate the nucleotide specificity of UCP1 inhibition using purified human UCP1 protein. We demonstrate that both purine and pyrimidine nucleotides can act as inhibitors, indicating that UCP1 has a broad inhibitor specificity. These observations fundamentally challenge the textbook notion that UCP1 is solely inhibited by purine nucleotides and provide new insights into our understanding of the physiological regulation of UCP1.

## Results

### Pyrimidine nucleotides bind and inhibit UCP1

The availability of purified human UCP1 provides a unique opportunity to revisit its nucleotide specificity. We have previously shown that the thermostability of proteins can be enhanced by the binding of ligands (Crichton et al, 2015; Harborne et al, 2020; Pyrihová et al, 2024; Mavridou et al, 2022; Majd et al, 2018) and can thus be used to identify interacting compounds. The classical UCP1 inhibitors, purine nucleotides, significantly increase the thermostability of UCP1 (Lee et al, 2015; Jones et al, 2023). To assess the binding of pyrimidine nucleotides to UCP1, differential scanning fluorimetry was used to determine the apparent melting temperature ($T_m$) of detergent-solubilised UCP1 with and without 1 mM nucleotide at low pH (pH 6.0) (Fig. 1B). For all tested nucleotides, a significant shift in apparent melting temperature was observed ($p < 0.0001$) in the presence of each nucleotide (Fig. 1B). The temperature shifts in the presence of 1 mM of nucleotide ($\Delta T_m$) were 19.6 °C for ATP, 18.9 °C for thymidine triphosphate (dTTP), 16.2 °C for GTP, 15.3 °C for UTP, and 12.5 °C for CTP (Fig. 1C). Notably, the pyrimidine nucleotide dTTP stabilised UCP1 significantly more than GTP ($p < 0.0001$) with a difference of 2.7 °C. dTTP differs from the other tested nucleotides, because the sugar moiety has a hydrogen rather than a hydroxyl group at the second position (Fig. 1A), but it is the only physiologically relevant thymidine nucleotide (Traut, 1994). Interestingly, the shift caused by the binding of the pyrimidine nucleotide UTP was not significantly different from that of the purine nucleotide GTP ($p = 0.2641$), whereas CTP, another pyrimidine nucleotide, had a significantly lower shift ($p < 0.001$). These results show that pyrimidine nucleotides are capable of stabilising UCP1 in thermostability assays.

The stabilisation of UCP1 by purine nucleotides has been shown to be pH-dependent (Echtay et al, 2018; Klingenberg, 1988; Jones et al, 2023; Lee et al, 2015). To assess whether the same pH-dependent interaction is observed for pyrimidine nucleotides, stabilisation by 1 mM nucleotide was tested in a range between pH 6.0 and 8.0. All nucleotides showed a pH-dependent interaction with UCP1 (Fig. 1D; Appendix Fig. S1), with the highest shift at pH 6.0, showing this property is independent of the nucleobase. This observation provides further evidence that the pH-dependence is not associated with nucleobase binding, but with binding of the phosphate groups, as they are involved in the proton-mediated bonding with the negatively charged residues of the matrix salt bridge network (Jones et al, 2024).

To understand the interactions between UCP1 and both purine and pyrimidine nucleotides further, isothermal titration calorimetry was carried out to determine their binding affinities (Fig. 2A–E; Appendix Fig. S2). All nucleotides were found to bind with low micromolar affinity to UCP1 with dissociation constants ($K_d$) of 0.81 μM for ATP, 1.17 μM for dTTP,

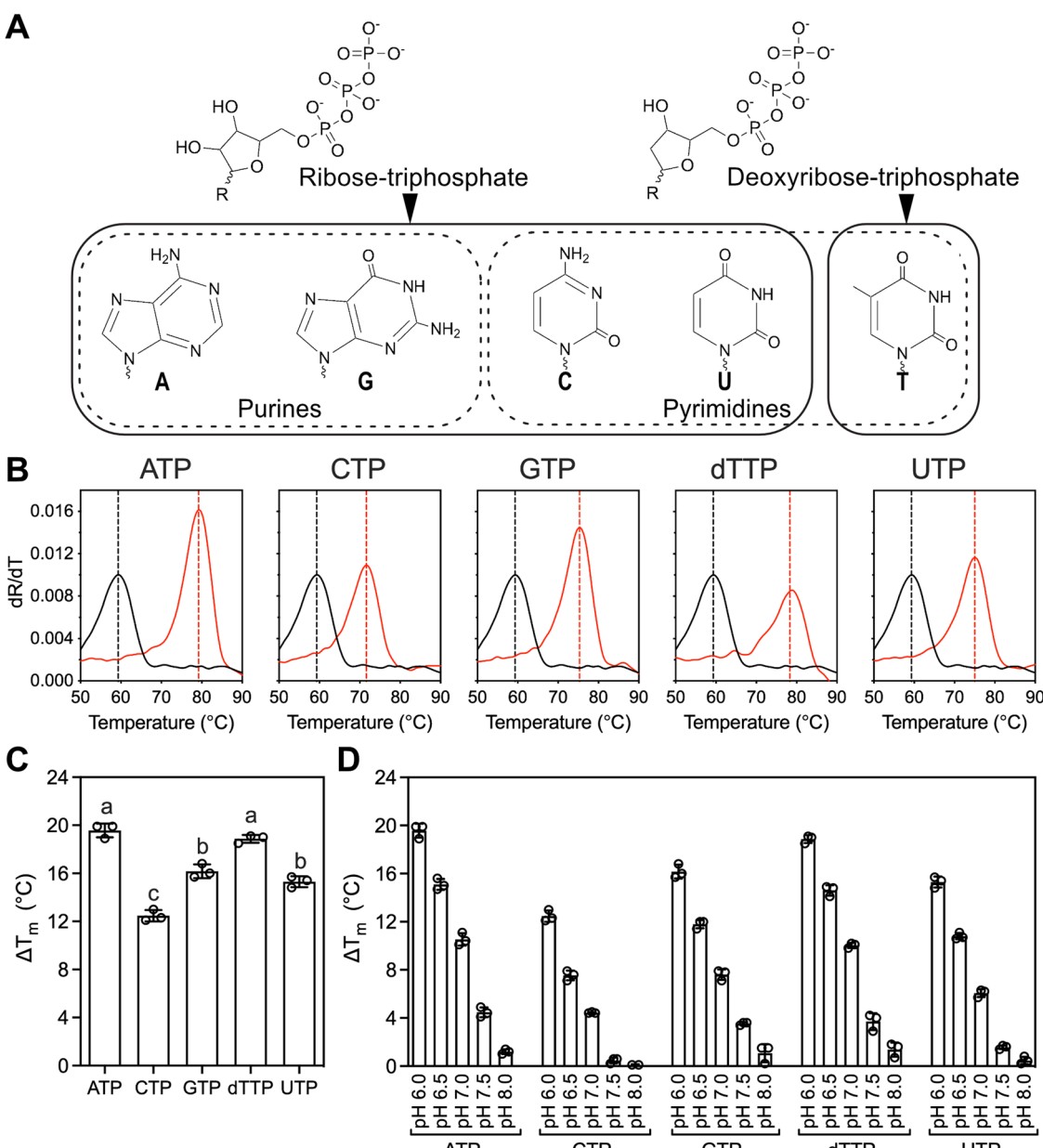

**Figure 1. Pyrimidine nucleotides bind uncoupling protein 1.**

(A) Chemical structures of the nucleotides. Adenine (A), guanine (G), cytosine (C), uridine (U) and thymine (T). Dotted lines show either purine or pyrimidine nucleobases. Solid lines separate ribose and deoxyribose containing nucleotides used in this study. (B) Binding of nucleotide to human UCP1 measured by thermal stability shift assays. The first derivative of the unfolding curve (dR/dT) with (red) and without (black) 1 mM nucleotide is shown with the apparent melting temperature ($T_m$) indicated by a dashed line. (C) The mean and standard deviation of three biological repeats of the thermal shift ($\Delta T_m$) caused by 1 mM nucleotide at pH 6.0. Nucleotides with significantly different $\Delta T_m$ are indicated by differing letters (one-way ANOVA with Tukey's post hoc test for multiple comparisons) (ATP vs. GTP, $p < 0.0001$; ATP vs. CTP, $p < 0.0001$; ATP vs. TTP, $p = 0.4466$; ATP vs. UTP, $p < 0.0001$; GTP vs. CTP, $p < 0.0001$; GTP vs. TTP, $p = 0.0004$; GTP vs. UTP, $p = 0.2641$; CTP vs. TTP, $p < 0.0001$; CTP vs. UTP, $p = 0.0002$; TTP vs. UTP, $p < 0.0001$). (D) The mean and standard deviation of three biological repeats of the thermal shift ($\Delta T_m$) caused by 1 mM nucleotide at pH 6.0–8.0. Source data are available online for this figure.

3.10 μM for UTP, 3.79 μM for GTP, and 5.22 μM for CTP (Fig. 2F). The binding affinity measured by isothermal calorimetry correlates inversely with the stabilisation achieved by the thermostability analysis ($R^2 = 0.903$) (Appendix Fig. S2F).

To assess whether pyrimidine nucleotides can also inhibit UCP1 activity, the protein was reconstituted into liposomes and the ability

of nucleotides to inhibit fatty acid-induced proton conductance was assessed fluorometrically (Lee et al, 2015; Kang and Chen, 2023) (Fig. 2G). We found that both purine and pyrimidine nucleotides inhibit oleic acid-induced proton conductance of UCP1 to a similar extent (Fig. 2H), significantly reducing proton influx ($p < 0.0001$).

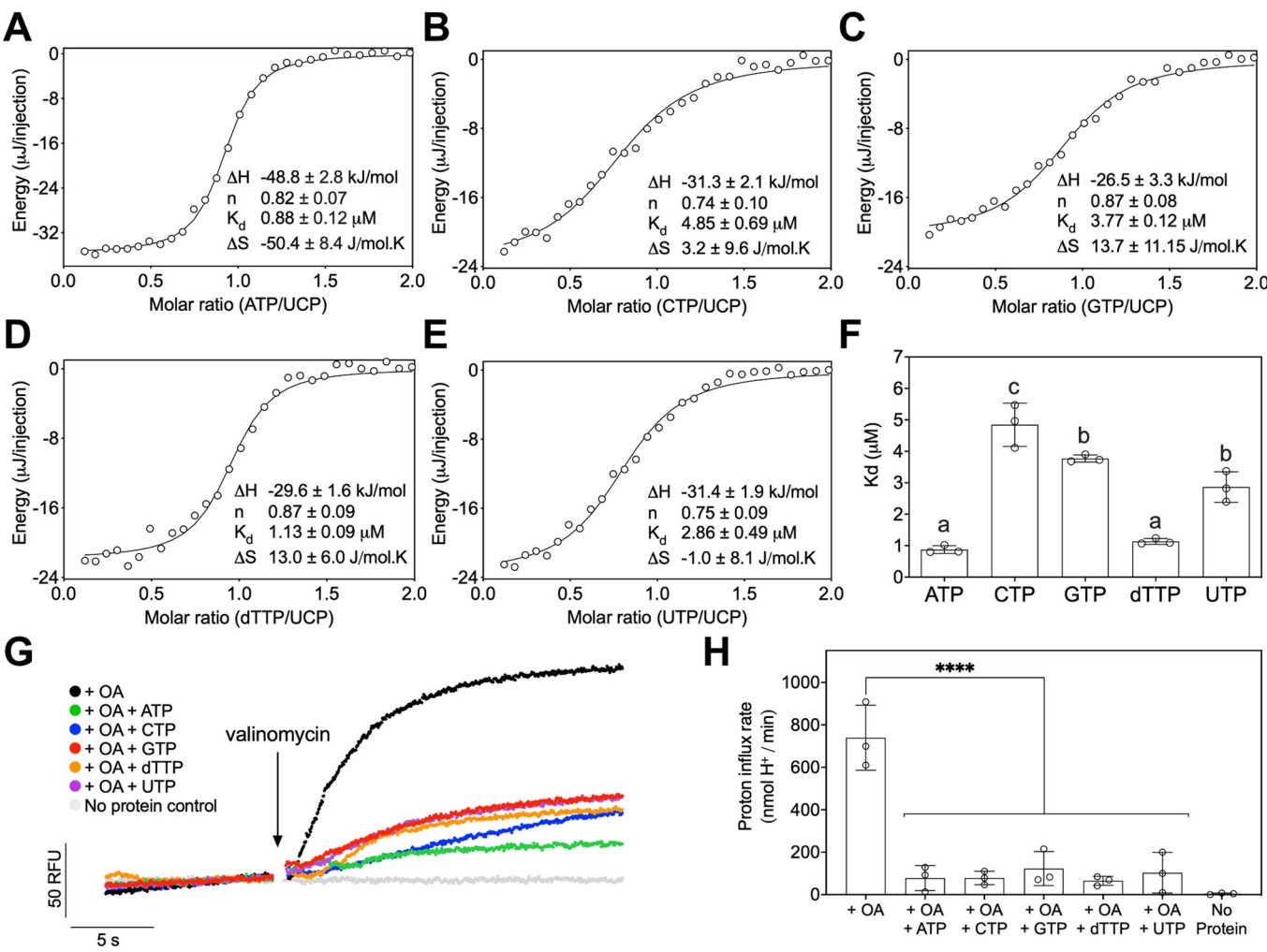

**Figure 2. Nucleotide specificity of uncoupling protein 1.**

(A–E) Representative isotherms from isothermal calorimetry fitted to a one-binding model to determine $\Delta H$, n, $K_d$ and $\Delta S$ values for each interaction. Mean ± standard deviation $\Delta H$, n, $K_d$ and $\Delta S$ values are presented for each nucleotide from three biological repeats. (F) Mean ± standard deviation $K_d$ of three biological repeats for each nucleotide. Nucleotides with significantly different binding $K_d$ are indicated by differing letters (one-way ANOVA with Tukey's post hoc test for multiple comparisons) (ATP vs. GTP, $p < 0.0001$; ATP vs. CTP, $p < 0.0001$; ATP vs. TTP, $p = 0.9205$; ATP vs. UTP, $p = 0.0007$; GTP vs. CTP, $p = 0.0413$; GTP vs. TTP, $p < 0.0001$; GTP vs. UTP, $p = 0.095$; CTP vs. TTP, $p < 0.0001$; CTP vs. UTP, $p = 0.0007$; TTP vs. UTP, $p = 0.0019$). (G) Representative traces showing proton uptake into reconstituted liposomes measured by pyranine fluorescence in the presence of 200 μM oleic acid (OA) and 500 μM nucleotide as indicated. Proton uptake was initiated by generating an electrochemical gradient using 2.5 μM valinomycin where indicated. (H) Mean ± standard deviation initial proton influx rate from three independent reconstitutions. ****$p < 0.0001$ for proton uptake with any given nucleotide compared with oleic acid alone (one-way ANOVA with Dunnett's test for multiple comparisons) (+OA vs. No Protein, $p < 0.0001$; +OA vs. +OA + ATP, $p < 0.0001$; +OA vs. +OA + GTP, $p < 0.0001$; +OA vs. +OA + CTP, $p < 0.0001$; +OA vs. +OA+dTTP, $p < 0.0001$; +OA vs. +OA + UTP, $p < 0.0001$). Source data are available online for this figure.

Together, our data show that UCP1 can bind both purine and pyrimidine nucleotides with affinities of physiological relevance. In addition, they show that pyrimidine nucleotides can functionally inhibit UCP1 proton conductance to a similar extent as purine nucleotides.

## The structure of UCP1 bound to UTP

To determine how pyrimidine nucleotides bind to UCP1, the UTP-bound structure was solved by cryo-EM (Appendix Table S1). Since UCP1 is conformationally dynamic, small (33 kDa) and has three-fold pseudosymmetry, nanobodies, modified to Pro-macrobodies (Botte et al, 2022), were used to aid the structural determination, as

in the case of the GTP-bound UCP1 structure (Jones et al, 2023). The density map, at a nominal resolution of 3.03 Å (Appendix Figs. S3, S4), enabled modelling of UCP1, the UTP nucleotide and partial modelling of three bound cardiolipin molecules (Fig. 3A). UTP interacts with UCP1 in a similar manner to purine nucleotides, explaining the similar binding and inhibitory properties. The phosphate group forms ionic interactions with K38 and K138 of the matrix network and the three arginine residues of the substrate binding site contact points ('the arginine triplet') (Fig. 3B). In addition, UTP forms hydrogen bond interactions between the α-phosphate and Q85 and the ribose group and R183 (Fig. 3B). Furthermore, R92 forms a cation-π interaction with the pyrimidine

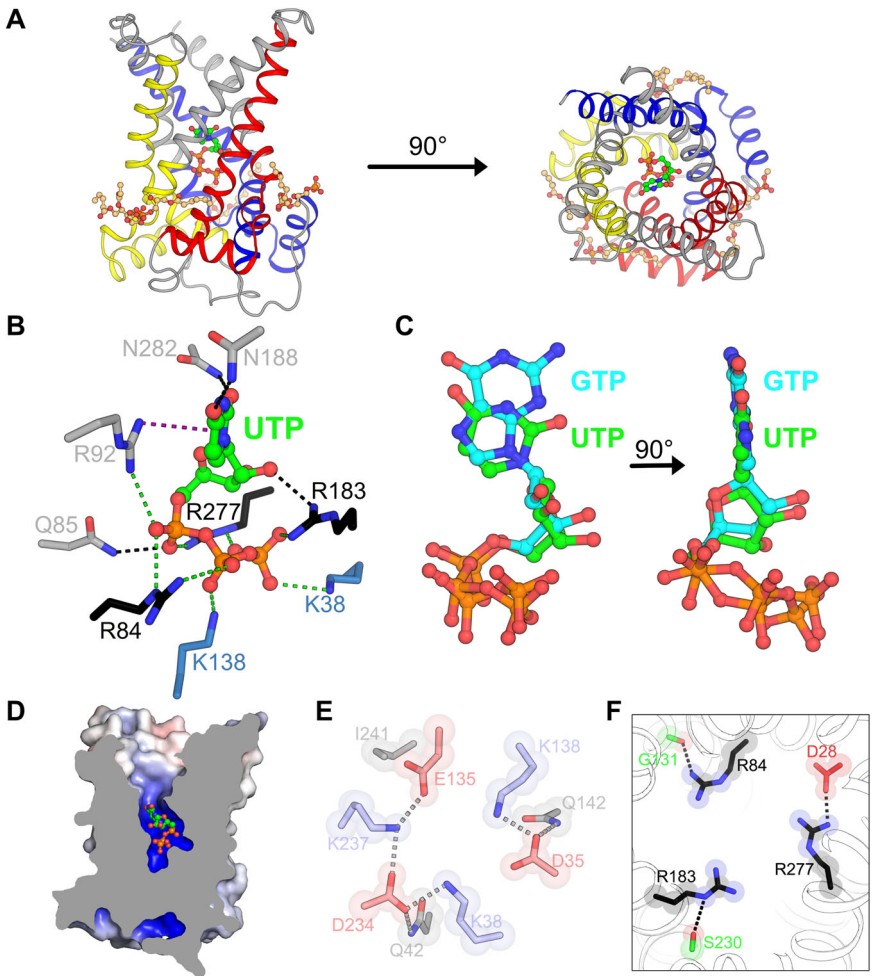

**Figure 3. UTP binding to uncoupling protein 1.**

(A) Structure of UCP1 with bound UTP (green) and three cardiolipin molecules (wheat). Core elements 1, 2, and 3 are coloured by domain in blue, yellow, and red, respectively, and the gate elements in grey. (B) UTP binding site. Residues are coloured by function: matrix network residues are shown in blue, arginine triplet residues are shown in black, and other residues involved in binding are shown in grey. Ionic interactions are shown with green broken lines, hydrogen bonds with black broken lines and the cation-π interaction with purple broken line. (C) Comparison of the binding pose between UTP (green) and GTP (cyan) (PDB:8G8W) (Jones et al, 2023) in UCP1. (D) Cross section through UCP1 showing the central water-filled cavity with UTP (green). The surface is coloured by the electrostatic potential (blue, +15 kT e$^{-1}$; white, neutral; red, −15 kT e$^{-1}$), calculated by ABPS. (E) Matrix network of UTP-bound UCP1. Positively charged residues are in blue, negatively charged residues are in red, and neutral residues are in grey. Broken lines show polar interactions. (F) Contact point residues of UCP1 inhibited with UTP. Black broken lines represent hydrogen bonds.

ring system, in a similar way as the purine rings of ATP and GTP (Fig. 3B) (Kang and Chen, 2023; Jones et al, 2023, 2024). The $O_4$ and $O_6$ of the uridine base also forms hydrogen bonds with N188 and N282, respectively (Fig. 3B), similar to the binding arrangement of the adenine moiety of ATP (Kang and Chen, 2023; Jones et al, 2024). Similarly, the $N_3$ of the guanine base of GTP interacts with N282, but the $N_2$ has an ionic interaction with E191 (Jones et al, 2023, 2024), which is absent in both the ATP and UTP bonding arrangements.

Comparison with the GTP structure shows that the UTP binding poses are almost identical, with the guanine and uridine rings partially overlapping (Fig. 3C). Furthermore, the UTP-bound structure also shows that the phosphate moieties are in the same position as those of GTP and ATP, where the negatively charged phosphate groups are within bonding distance of the negatively charged residues of the matrix network. This binding arrangement

requires proton-mediated bonds, explaining the pH-dependent binding observed for all nucleotides (Jones et al, 2024). Together, the nucleotide-bound structures suggest that all nucleotides bind in the same position, despite some interaction differences at the nucleobase.

The binding of UTP creates a proton-impermeable barrier between the intermembrane space and the mitochondrial membrane (Fig. 3D). The barrier is generated by the glutamine brace interactions, engaged matrix salt bridge network and C-terminal ends of the odd-numbered helices. The matrix network is partially distorted due to the direct interaction of residues (K38 and K138) with the phosphate moieties of UTP, consisting of both inter- and intra-domain bonds (Fig. 3E). This distortion is similar to the one observed in the purine nucleotide-bound structures and explains how the nucleotide prevents opening of the network, maintaining a proton-impermeable state. The UTP-bound structure confirms that

pyrimidine nucleotides inhibit UCP1 by the same mechanism as purine nucleotides, by locking the matrix network, by generating a proton-impermeable layer and by preventing conformational changes (Fig. 3D,E) (Jones et al, 2024, 2023).

All resolved structures of UCP1 to date have failed to identify a proton-conductance pathway (Kang and Chen, 2023; Jones et al, 2023). However, one of the few specific adaptations in the central cavity of UCP1 when compared with other SLC25 family members is D28, which has been shown to be important for proton conductance (Echtay et al, 2000; Robinson et al, 2008). In the purine nucleotide-bound states D28 was found in an ionic bond with R277, one of the contact point arginine residues (Jones et al, 2024). However, in the nucleotide-free state (Kang and Chen, 2023), this bond was broken, potentially enabling D28 to bind a proton as part of the proton conductance mechanism (Jones et al, 2023, 2024). Here, in the UTP-bound structure, we find the same ionic bond between D28 and R277, revealing the same potential mechanism of inhibition (Fig. 3F).

Thus, the UTP-bound structure of UCP1 shows that there are many similarities between the molecular mechanism of pyrimidine and purine nucleotide binding, explaining the comparable binding affinities and levels of proton conductance inhibition.

### UCP1 has evolved to have a limited nucleotide specificity

UCP1 is phylogenetically related to mitochondrial dicarboxylate carriers, gaining thermogenic properties as a later evolutionary development (Hughes et al, 2009; Robinson et al, 2008). The mitochondrial dicarboxylate carrier (DIC, SLC25A10) and oxoglutarate carrier (OGC, SLC25A11) are most closely related to UCP1 (Gaudry et al, 2019; Hughes et al, 2009) and share many residues involved in nucleotide binding in UCP1, such as the arginine triplet at the contact points, R92 and Q85 (Jones et al, 2024). Given the similarity of the central cavity of UCP1, OGC and DIC, we tested whether the dicarboxylate carriers have the ability to bind nucleotides. For this purpose, we purified human DIC and OGC, and tested whether nucleotides were able to generate a thermostability shift, as observed for their physiological substrates (Pyrihová et al, 2024). However, all tested nucleotides caused a negligible shift with both DIC or OGC at all pH values tested between 6.0 and 8.0 (Fig. 4A; Appendix Fig. S5). Thus, despite having many residues in common (Fig. 4B–D), UCP1 has evolved to have a few specific residues that allow high-affinity binding of nucleotides without a strong selection for nucleobase.

While most of the residues that are involved in the nucleotide interactions with UCP1 are also observed in DIC and OGC, the matrix network residues differ (Fig. 4C,D). Both DIC and OGC have weaker networks due to the replacement of the positively charged lysine (K138) with asparagine in DIC (Fig. 4C) and leucine in OGC (Fig. 4D). This observation suggests that K138 is an important residue for nucleotide binding. Another adaptation specific to uncoupling proteins is W281 (Fig. 4B), which is a histidine residue in both DIC and OGC (Fig. 4C,D) and could be important for nucleotide binding (Gagelin et al, 2023), as it is located within van der Waals distance of both the purine (Jones et al, 2023; Kang and Chen, 2023) and pyrimidine rings (Fig. 4B) in the nucleotide-bound state of UCP1. In DIC, N188 and N282, which in UCP1 are involved in the binding of nucleobases, are replaced by threonine residues (Fig. 4C), while in OGC, only N282

is substituted with a threonine residue (Fig. 4D). In addition, in both DIC and OGC, UCP1 residue I187 is replaced with a valine residue (Fig. 4B–D).

## Discussion

It is widely believed that UCP1 inhibition is specific to purine nucleotides. Here we have shown that pyrimidine nucleotides also bind to UCP1 with similar affinities and inhibit proton conductance, just like purine nucleotides (Fig. 2F,H). Structural analysis of the UTP-bound state of UCP1 demonstrates that pyrimidine nucleotides interact with UCP1 through bonding interactions similar to those observed for purine nucleotides (Fig. 3). UTP binding also generates the same proton impermeable state as purine nucleotides (Fig. 3D–F) (Jones et al, 2023, 2024; Kang and Chen, 2023), indicating that pyrimidine nucleotides inhibit proton conductance with the same mechanism. The availability of well-folded, pure human UCP1 protein was a key factor in establishing the broad inhibitory specificity. We show that human UCP1 can bind any nucleotide regardless of nucleobase or the functional group at the $C_2$ position of the ribose, i.e. H or OH (Fig. 1A).

Using the UTP-bound structure as a template we can also rationalise how CTP and dTTP bind and inhibit UCP1 (Appendix Fig. S6). CTP differs from UTP at the $C_4$ position of the nucleobase, where cytosine has an amine group and uridine a carbonyl group. The UTP structure shows that this $C_4$ carbonyl has a hydrogen bond with N188. A simple change of rotamer allows N188 to form a hydrogen bond interaction with the amine group of cytosine (Appendix Fig. S6A). The thymine and uracil nucleobases differ only at the $C_5$ position of the pyrimidine ring, with thymine having a methyl group rather than a hydrogen atom. Modelling of dTTP indicates that the methyl group fits into the binding site without steric hindrance, therefore the thymine base can occupy the same binding position as the uridine base with similar interactions to UCP1. The nucleotide dTTP also differs because it lacks a ribose hydroxyl group and will thus lose a hydrogen bond with R183 (Appendix Fig. S6B), observed in all the nucleotide-bound structures (Fig. 3A) (Jones et al, 2024). However, the other hydroxyl group of the ribose ring of dTTP could be within bonding distance of R277, which could compensate for the lost interaction (Appendix Fig. S6B).

This study highlights the differences in the binding affinities of the various nucleotides with UCP1 (Fig. 2F). The nucleotides can be divided into three significantly different classes: (i) ATP and dTTP have the highest binding affinities, (ii) GTP and UTP have intermediary affinities, whereas (iii) CTP has the lowest affinity. Interestingly, ATP binding to UCP1, as measured by isothermal titration calorimetry, had a much greater and more negative enthalpy and entropy change than all other nucleotides (Fig. 2A). However, when comparing the ATP-bound structure (Kang and Chen, 2023) with the UTP-bound structure, solved here (Fig. 3), we find no differences in the polar interactions, suggesting that they alone are insufficient to explain the difference in measured affinities. Thymine and adenine are the most hydrophobic nucleobases compared to the more hydrophilic guanine, uracil and cytosine (Shih et al, 1998), which correlates with the binding affinities of the nucleotides (Fig. 2F). Even though the binding site

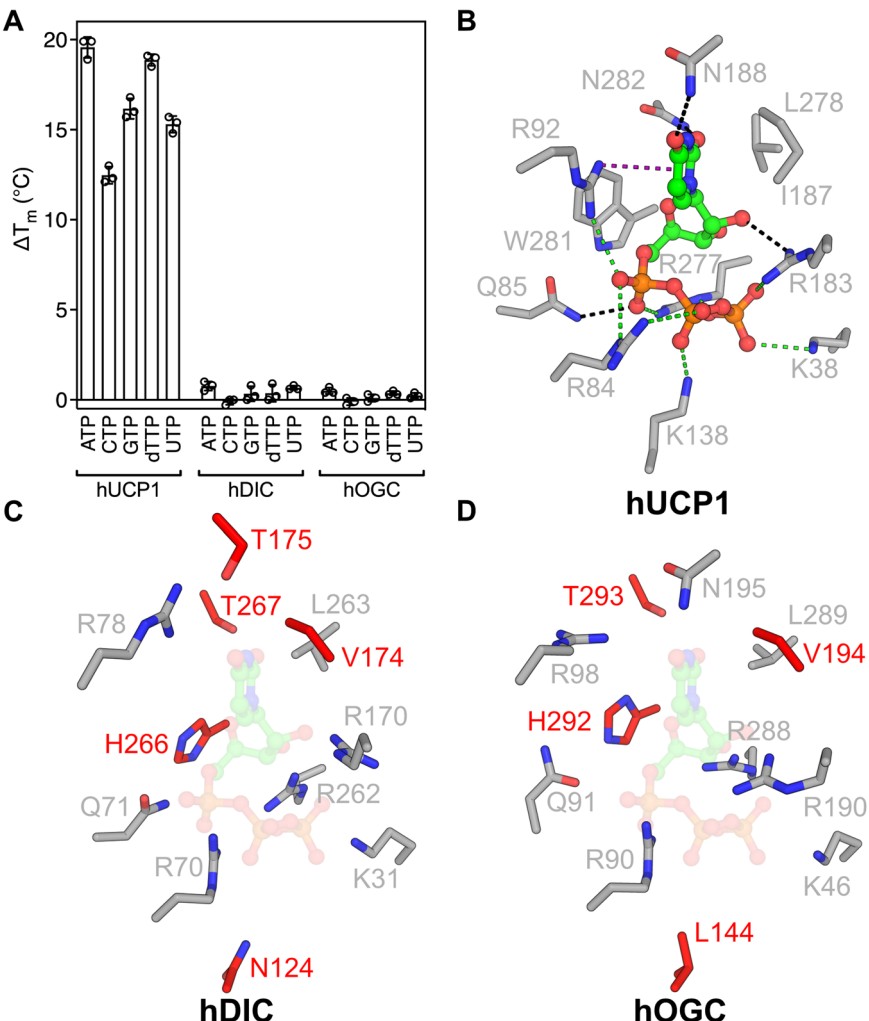

**Figure 4. UCP1-related mitochondrial carriers OGC and DIC do not bind nucleotides.**

(A) Change in apparent melting temperature ($\Delta T_m$) of hUCP1, hDIC and hOGC with the addition of 1 mM nucleotide at pH 6.0. (B) Residues within van der Waals distance of UTP (determined using LigPlot (Laskowski and Swindells, 2011)) are shown for the UTP-bound structure. (C) AlphaFold3 (Abramson et al, 2024) model of hDIC (D) AlphaFold3 (Abramson et al, 2024) model of hOGC. In (C) and (D), outline of UTP based on the binding in UCP1 for comparison purposes. Residues in red are modifications from the central binding site of UCP1. Source data are available online for this figure.

is non-selective, this correlation suggests that nucleobases bind more tightly to the binding site when they are hydrophobic. The hydrophobic W281, I187, and L278 form van der Waals stacking arrangements with the nucleobase in the binding site and could therefore be responsible for the observed hydrophobic effect (Fig. 4B).

The inhibition of UCP1 activity is an important regulator of non-shivering thermogenesis and prevents uncoupling of the mitochondria without activation. With the exception of dTTP, the measured affinities are orders of magnitude higher than the average cytoplasmic concentrations measured in human cells: 2.5 mM for ATP, 0.23 mM for GTP, 0.23 mM for UTP, 83 μM for CTP and 5.4 μM for dTTP (Traut, 1994). Given the millimolar concentration of ATP and ADP in the cytoplasm (Traut, 1994), these nucleotides are likely to be the most relevant inhibitors of UCP1 under physiological conditions. However, our results show

there is no selectivity for nucleobase and previous work has indicated that both diphosphate and triphosphate purine nucleotides inhibit UCP1 (Jones et al, 2023; Lee et al, 2015; Echtay et al, 2018; Heaton and Nicholls, 1977), therefore, it is likely that any diphosphate or triphosphate nucleotide will inhibit UCP1. As GTP and UTP have similar cytoplasmic concentrations (Traut, 1994) and binding affinities (Fig. 2F), both could inhibit UCP1 proton influx (Fig. 2H) and thus play equivalent roles in the regulation of UCP1 function. Based on the different concentrations of nucleotides in the cytosol (Traut, 1994), the regulation of UCP1 would still be dominated by adenine nucleotides, but about 1 in 5 of the inhibition events will involve other nucleotides. However, it is worth noting that the concentrations of the various nucleotides in the intermembrane space are unknown and subject to continuous flux due to enzymatic reactions and transport events into and out of the mitochondrion in a condition-dependent manner. Thus, the

physiological role of the nucleotides in UCP1 inhibition is not fully understood and these processes may account for the differences observed in UCP1 inhibition studies carried out with whole mitochondria.

UCP1 is a relatively late evolutionary development, mostly associated with mammals, and likely evolved from the closely related DIC and OGC carriers (Gaudry et al, 2019; Hughes et al, 2009), which transport small metabolites across the mitochondrial inner membrane, unlike UCP1 (Jones et al, 2024). Recently, it has been claimed that OGC shows proton conductance that can be inhibited by nucleotides (Žuna et al, 2024). However, our study using folded and functionally active OGC (Pyrihová et al, 2024), conclusively shows that neither OGC nor DIC can bind nucleotides in the same way as UCP1 (Fig. 4A; Appendix Fig. S6). If inhibition of dicarboxylate solute transport by nucleotides were possible, it would interfere with their crucial function in mitochondrial energy metabolism.

The activation mechanism of UCP1 by fatty acids and its interplay with nucleotide inhibition remain unresolved on the molecular level (Jones et al, 2024; Nicholls, 2021). One of the key questions is how UCP1 overcomes continuous inhibition by nucleotides under physiological conditions, given their low dissociation constants. Our study adds to this conundrum, as we now show that pyrimidine nucleotides can also act as inhibitors of UCP1, thus increasing the number of inhibitors UCP1 must overcome to be activated.

It has been postulated that an increase in cytosolic $Ca^{2+}$, following adrenergic stimulation of BAT (Leaver and Pappone, 2002), may act to sequester nucleotides, thus terminating inhibition and allowing UCP1 activation (Bast-Habersbrunner and Fromme, 2020). Indeed, we tested whether the presence of different concentrations of $Ca^{2+}$ or $Mg^{2+}$ altered ATP or UTP binding to human UCP1 and found no difference for either cation, until supramaximal concentrations were added (Appendix Fig. S7). The structures of nucleotide-inhibited UCP1, elucidated here and previously (Jones et al, 2023; Kang and Chen, 2023), show that the central cavity contains six positively charged residues involved in nucleotide binding (K38, R84, R92, K138, R183, R277), which can effectively compete with divalent cations for binding of the phosphate groups. Therefore, it is highly unlikely that the increase in cytosolic $Ca^{2+}$ concentrations, following adrenergic stimulation, plays a role in terminating nucleotide inhibition.

Our study clearly demonstrates that all nucleotides bind to and inhibit UCP1 proton conductance, rejecting the commonly accepted notion that purine nucleotides are the sole inhibitors. While there are subtle differences in binding affinities, the central cavity of UCP1 can bind all nucleotides regardless of the nucleobase, which has been explained here on the molecular level. This work shows that few adaptations were required to allow binding of the regulatory nucleotides with a mechanism that exploits their common chemical properties, in particular their proton-mediated binding via the phosphate moieties. Given the interest in harnessing UCP1 activity in BAT for treatment of metabolic diseases (Chondronikola et al, 2014; Nedergaard et al, 2007; Saito et al, 2009; Cypess et al, 2009), our study completes our understanding of UCP1 inhibition by naturally occurring nucleotides.

# Methods

**Reagents and tools table**

| Reagent/Resource | Reference or Source | Identifier or Catalog Number |
|---|---|---|
| **Experimental models** | | |
| *E. coli* strain MC1061 | ATCC | ATCC number: 47035 |
| *Saccharomyces cerevisiae* W303-1B | ATCC | ATCC number:201238 |
| **Recombinant DNA** | | |
| pYES3 vector | Invitrogen | V825220 |
| pBXNPHM3 vector | Addgene | ATCC number: 110099 |
| **Antibodies** | | |
| N/A | | |
| **Oligonucleotides and other sequence-based reagents** | | |
| **Chemicals, Enzymes and other reagents** | | |
| Dodecyl maltose neopentyl glycol | Anatrace | Cat#NG310 |
| Complete Mini EDTA-free protease inhibitor tablets | Roche | Cat#05056489001 |
| DNAse I | Roche | Cat#10104159001 |
| Nickel Sepharose (High Performance) | GE Healthcare | Cat#17526802 |
| TCEP | Generon | Cat#GEN-TCEP-10 |
| GTP | Merck | Cat#51120 |
| ATP | Merck | Cat#A1852 |
| dTTP | Merck | Cat#T0251 |
| UTP | Merck | Cat#94370 |
| CTP | Merck | Cat#C1506 |
| Valinomycin | Merck | Cat# 94675 |
| Oleic acid | Merck | Cat#O1008 |
| Sodium phosphate | Merck | Cat#342483 |
| Carbonyl cyanide 3-chlorophenylhydrazone | Merck | Cat#C2759 |
| Nickel-NTA | Qiagen | Cat#30430 |
| Sephadex G75 | GE Healthcare | Cat#17005001 |
| Strep-Tactin XT 4Flow | IBA | Cat#25030025 |
| Thiopropyl agarose 6B resin | Cytiva Lifesciences | Cat#17042001 |
| Factor Xa protease | NEB | Cat#P8010L |
| 3C protease | Merck | Cat# 71493 |
| Tetraoleoyl cardiolipin (TOCL) | Avanti Polar Lipids | Cat#710335 |
| **Software** | | |
| Coot | (Emsley et al, 2010) | https://www2.mrc-lmb.cam.ac.uk/personal/pemsley/coot/ |
| CryoSPARCv4.0.3 | (Punjani et al, 2017) | https://cryosparc.com |

| Reagent/Resource | Reference or Source | Identifier or Catalog Number |
|---|---|---|
| Molprobity | (Chen et al, 2010) | http://molprobity.biochem.duke.edu/ |
| Pymol | Schrödinger | https://pymol.org/2/ |
| UCSF Chimera | (Pettersen et al, 2004) | https://www.cgl.ucsf.edu/chimera/ |
| Other | | |

## Expression and purification of mitochondrial carriers

Sequences encoding human UCP1 (Uniprot: P25874), OGC (Uniprot: Q02978) and DIC (Uniprot: Q9UBX3) were modified to encode an upstream N-terminal His$_8$ tag and Factor Xa protease cleavage site (Pyrihová et al, 2024; Jones et al, 2023). The sequences were cloned into a pYES2/CT vector and expressed in either *Saccharomyces cerevisiae* strain W303.1B (UCP1 and OGC) or the protease-deficient strain BJ2186 (DIC). Positive transformants were selected on Sc-Ura + 2% (*w/v*) glucose plates. Yeast expressing UCP1 were grown in 2 L of SP media (0.67% YNB, 0.1% KH$_2$PO$_4$, 0.12% (NH$_4$)$_2$SO$_4$, 0.1% casamino acids, 20 mg/L L-tryptophan, 40 mg/L adenine, 0.1% glucose, 2.0% lactic acid, pH 5.5) at 30 °C for 24 h. Yeast expressing DIC and OGC were grown in Sc-Ura + 2% glucose. Pre-cultures were inoculated into 100 L of either yeast peptone (YP) media supplemented with 3% lactic acid (UCP1) or 3% glycerol+0.1% glucose (DIC and OGC) in an Applikon 140 Pilot System with an eZ controller. Cells were grown at 30 °C for 20 h, induced with 0.4% galactose for 4 h and harvested by centrifugation (4000 × *g*, 20 min, 4 °C). Mitochondria were prepared by disrupting cells with a bead mill (Dyno-Mill Multilab, Willy A. Bachofen AG Maschinenfabrik, Switzerland), as previously described (King and Kunji, 2020). The total mitochondrial protein concentration was adjusted to 20 mg/mL with 0.1 M Tris pH 7.4, 10% glycerol. Mitochondria were flash frozen in liquid nitrogen, and stored at −70 °C. Mitochondria (~1 g) were solubilized in 1% lauryl maltose neopentyl glycol (Anatrace), with 20 mM imidazole, 150 mM NaCl, 1 mM TCEP (Generon) and one cOmplete EDTA-free protease inhibitor tablet (Roche) for 1 h at 4 °C. The solubilisate was clarified by centrifugation (142,000 × *g*, 45 min, 4 °C) before batch binding for 1 h to 0.7 mL Ni Sepharose High Performance resin (GE Healthcare). The resin was washed with 20 mL of buffer A (20 mM Tris pH 7.4 (UCP1) or 20 mM HEPES pH 7.0 (DIC and OGC), 150 mM NaCl, 40 mM imidazole, 0.1 mg/mL tetraoleoyl cardiolipin (Avanti Polar Lipids), 0.1% lauryl maltose neopentyl glycol, 1 mM TCEP (for UCP1 purifications only)), followed by 5 mL of buffer B (20 mM Tris pH 7.4, 50 mM NaCl, 0.1 mg/mL tetraoleoyl cardiolipin, 0.1% lauryl maltose neopentyl glycol, 1 mM TCEP), under gravity flow. The Ni Sepharose was recovered as a slurry (total volume ≈ 1.0 mL), supplemented with 5 mM CaCl$_2$, 10 mM imidazole and 20 µg Factor Xa protease (NEB), and incubated overnight at 10 °C. The slurry was transferred into empty Proteus 1-Step Batch Mini Spin columns (Generon) and the protein was eluted from the resin by centrifugation (500 × *g*, 5 min, 4 °C). The protein concentration was determined by BCA assay (ThermoFisher) for UCP1 or spectrometry (NanoDrop Technologies) at 280 nm using calculated extinction coefficients for OGC and DIC (OGC: 29,000 M$^{-1}$ cm$^{-1}$; DIC: 20,900 M$^{-1}$ cm$^{-1}$).

## Differential scanning fluorimetry

Differential scanning fluorimetry (nanoDSF) (Alexandrov et al, 2008) was carried out with 2 µg UCP1, DIC or OGC protein (100 mM MES/HEPES, pH range 6.0–8.0, 25 mM NaCl, 1 mM TCEP, 0.01% lauryl maltose neopentyl glycol and 0.01 mg/mL tetraoleoyl cardiolipin) and, when required, 1 mM nucleotide, as indicated. The proteins have the following number of tryptophans: UCP1 (two); OGC (two); DIC (one). The samples were loaded into nanoDSF-grade standard glass capillaries. The temperature was increased by 4 °C every minute from 25 to 95 °C, the intrinsic fluorescence was measured in a Prometheus NT.48 nanoDSF device, and the apparent T$_m$ was calculated with the PR.Therm-Control software (NanoTemper Technologies).

## Isothermal titration calorimetry

Nucleotide binding to UCP1 was measured using a NanoITC-LV isothermal titration calorimeter (TA Instruments) at 20 °C (Lee et al, 2015). Purified human UCP1 was exchanged into ITC buffer (50 mM cacodylate pH 6.0, 0.01% lauryl maltose neopentyl glycol, 0.01 mg/mL tetraoleoyl cardiolipin, 1 mM TCEP) using a minitrap G-25 column. Both nucleotide titrant and protein samples were degassed under vacuum for 20 min before titration. Each nucleotide (750 µM) was titrated into between 70 and 83 µM UCP1 in 1 µL injections at 3-min intervals with a stirrer speed of 250 rpm. Isotherms were analysed by using the instrument software (NanoAnalyze) and fitted to a one-site binding model with ΔH, K$_d$, and stoichiometry as fitting parameters.

## Liposome reconstitution and proton conductance assay

Purified UCP1 was reconstituted into liposomes using 10 mg 1,2-dioleoyl-sn-glycero-3-phosphocholine (DOPC) supplemented with 0.5 mg 18:1 cardiolipin (TOCL). The lipids were dried under a stream of nitrogen before being rehydrated in 100 mM potassium phosphate pH 7.5, 0.2 mM EDTA. The lipids were solubilised using 2.5% pentaethylene glycol monodecyl ether (C$_{10}$E$_5$). 100 µg of purified UCP1 was added to the sample and the detergent was removed with SM-2 bio-beads with four additions of 30 mg every 20 min and a final addition of 240 mg before incubating overnight at 4 °C. Bio-beads were removed using empty micro-bio spins columns before the liposomes were exchanged into external buffer (280 mM sucrose, 0.5 mM HEPES, pH 7.5) using a PD-10 column. Proton uptake into liposomes was measured using an established UCP1 activity assay (Lee et al, 2015; Kang and Chen, 2023). Proteoliposomes (200 µL) were diluted to 1 mL using external buffer before the pH-sensitive fluorophore pyranine (0.5 µM) was added (from 1 mM stock, pH adjusted to 8.2 with TEA-OH) in a quartz cuvette with 200 µM oleic acid (including 1.6 mM methyl-β-cyclodextrin) and 500 µM nucleotide added from stocks, where indicated. Changes in pH were measured in a Shimadzu RF3501PC spectrofluorometer at 25 °C, with constant stirring (λ$_{ex}$ 467 nm, λ$_{em}$ 510 nm). A stable signal was recorded for 10 s before a membrane potential was induced through the addition of 2.5 µM valinomycin to drive proton uptake. Initial rates of proton uptake were estimated from fits of the valinomycin-induced progress curve using the appropriate exponential function ("plateau and one phase association"; GraphPad Prism software v10.2). Fluorescent signal

was calibrated to proton concentration in external buffer containing 0.5 μM pyranine, through sequential additions of 0.1 M $H_2SO_4$ or 0.1 M NaOH.

## Statistical analysis

The difference between thermostability apparent melting temperatures ($T_m$) and between dissociation constant measurements ($K_d$) were evaluated using one-way ANOVA with the Tukey test to correct for multiple comparisons. The difference in proton influx between oleic acid stimulated liposomes with and without nucleotide inhibition was evaluated by using a one-way ANOVA with correction for multiple comparisons carried out using the Dunnett test. Statistical analysis of the data was performed with GraphPad Prism.

## Cryo-EM sample preparation

Nanobodies against GDP-inhibited ovine UCP1 were generated as previously described (Jones et al, 2023). One llama (*Lama glama*) was immunized with UCP1 reconstituted in 1,2-dioleoyl-sn-glycero-3-phosphocholine:tetraoleoyl cardiolipin using established methods (Ruprecht et al, 2019; Lee et al, 2015). A phage display library of nanobodies was prepared in the pMESy4 vector from peripheral blood lymphocytes as described (Pardon et al, 2014). Nanobodies were identified by selecting phages that bound to solid-phase immobilized proteoliposomes in the presence of GDP and confirming lack of binding to empty liposomes. This work was done in compliance with both the European legislation (EU directive 2010/63/EC) and the Belgian Royal Decree of 29 May 2013 concerning the protection of laboratory animals with the exception that the animals are not specifically bred for such use.

Pro-macrobody (PMb) purifications were conducted following previously described methods (Botte et al, 2022; Jones et al, 2023). *Escherichia coli* MC1061 cells were transformed with the pBXNPHM3 vector (Addgene: 110099) expressing PMb sequences for CA9871 and CA9865. The bacteria were cultured in 2 L of terrific broth (supplemented with 100 μg/mL ampicillin, 0.1% glucose, 1 mM $MgCl_2$, 1.0% glycerol) at 37 °C until an $OD_{600}$ of 0.7 upon which 0.02% arabinose was used to induce expression for 3.5 h. The bacteria were then harvested and resuspended in 150 mM NaCl, 50 mM Tris-HCl pH 8, 20 mM imidazole, 5 mM $MgCl_2$, 10% glycerol, deoxyribonuclease I (10 μg/mL), and cOmplete Mini EDTA-free protease inhibitor tablet (Roche). The bacteria were lysed using a cell disruptor (Constant Cell Disruption Systems) at 30 kpsi and centrifuged (205,000 × g, 30 min, 4 °C). The supernatant was incubated with 3 mL of Ni–nitrilotriacetic acid slurry for 1 h. The resin was washed with 150 mM KCl, 40 mM imidazole (pH 7.5), and 10% glycerol. The protein was eluted with 150 mM KCl, 300 mM imidazole (pH 7.5), and 10% glycerol. The elute was desalted into 150 mM KCl, 20 mM imidazole (pH 7.5), and 10% glycerol using a PD-10 column. The protein was cleaved overnight with 3C protease (Merck) to remove N-terminal MBP. The His-tagged MBP was removed by 0.7 mL of Ni–nitrilotriacetic acid slurry for 1 h. To further purify the protein was bound to 750 μL of Strep-Tactin XT 4Flow slurry (IBA) for 1 h and eluted with 50 mM biotin.

The purified PMbs and UCP1 were exchanged into 20 mM MES (pH 6.0), 150 mM NaCl, 0.02% decyl maltose neopentyl glycol,

0.02 mg/mL tetraoleoyl cardiolipin, and 1 mM TCEP using a PD-10 column. The protein concentrations were determined by BCA assay (Thermo Fisher Scientific). UCP1, PMb65, and PMb71 were mixed stoichiometrically and incubated overnight at 4 °C. To separate the complex from unbound PMbs the mixture was subjected to size exclusion chromatography using a Superdex-200 10/300 GL column using SEC buffer (20 mM MES pH 6.0, 150 mM NaCl, 0.02% lauryl maltose neopentyl glycol, 0.02 mg/mL tetraoleoyl cardiolipin, 1 mM TCEP). The peak containing the complex was collected and concentrated to 3.0 mg/mL. The protein was supplemented with 2 mM UTP and 2 mM D-maltose to generate the closed conformation of the maltose binding protein before freezing.

## Cryo-EM data collection and processing

Quantifoil R1.2/1.3 300-mesh holey carbon copper grids were glow discharged for 30 s at 10 mA before sample freezing. UCP1:macro-body complex (3 μL) supplemented with 0.05% fluorinated octyl maltoside at a concentration of 3 mg/mL was applied to the grid (blot time, 2 s; blot force, −5) and the grids were plunge-frozen in liquid ethane cooled with liquid nitrogen using a Vitrobot Mark IV operated at 4 °C and under 95% humidity. The UCP1 datasets were collected at the UK National Electron Bio-Imaging Centre (eBIC) and from two grids each using a Gatan K3 detector mounted on a FEI Titan Krios transmission electron microscope with a 100 μm objective aperture and 70 μm C2 aperture. Data were collected using EPU (Mastronarde, 2005) in super-resolution mode at 0.645 Å/pixel (nominal magnification 130,000×) with defocus range of −1.8 to −0.6 μm in 0.2 μm increments and the autofocus routine run every 10 μm. The total dose of 50 $e^-$/Å$^2$ with 1.3 s exposure. Data were acquired as five shots per hole using the aberration-free image shift (AFIS) mode. Cryo-EM data processing of UCP1 was performed in Cryo-SPARCv3.3.2 (Punjani et al, 2020) (Appendix Fig. S3). A total of 39,450 movies were collected. Beam-induced motion correction was performed using Patch Motion and contrast transfer function (CTF) estimation was done using Patch CTF. Template picker was used to pick particles and then curated to a total of 12,302,465 particles. The particles were binned to 2.5 Å/pixel during extraction and subjected to multiple rounds of two-dimensional classification. A total of 1,506,075 particles were further classified into four classes using cryoSPARC ab initio reconstruction (default settings). This separated UCP1 bound with two PMbs (cytoplasmic and mitochondrial matrix side) from junk classes comprising of free-detergent micelles and UCP1 with one or no PMb bound. The good class obtained from the six-class ab initio reconstruction was further cleaned by performing two further rounds of heterogenous refinement with one good reference, corresponding to the UCP1-PMb65-PMb71 complex, and two bad references of each junk class. Then non-uniform refinement on the 708,661 good particles using the good volume from the initial heterogenous refinement yielded a reconstruction at 3.27 Å resolution. Reference Based Motion Correction (RBMC - cryoS-PARC v4.5.3) was applied to the curated particle set using default parameters. Local refinement using the polished particles from the RBMC job, the volume from the non-uniform reconstruction, and a mask was performed, leading to a final reconstruction at a gold-standard FSC resolution of 3.03 Å. Default parameters were used with the following exceptions: use pose/shift Gaussian prior during

alignment, standard deviation (deg) 6, standard deviation (Å) 2, with an initial low-pass resolution of 8 Å. The mask used for the local refinement was created using Chimera 1.13.1. The map was filtered using vop gaussian (sdev 4), and the threshold was adjusted to encompass the carrier and the nanobody, excluding the Maltose Binding Protein (Appendix Fig. S3).

## Modelling

Model building was initiated by placing UCP1 + GTP structure (PDB:8G8W) (Jones et al, 2023) as a rigid-body into the cryo-EM map using ChimeraX (Meng et al, 2023). The structure was re-built using Coot (Emsley et al, 2010) and Isolde (Croll, 2018) run via ChimeraX (Meng et al, 2023) (Appendix Fig. S4). Clear density for UTP was visible in the UCP1 cavity and the ligand was fitted into the cryo-EM map. Density was also apparent for three cardiolipin molecules associated with UCP1, and partial models were fitted into the map, where supported by the density. Restraint files for UTP and cardiolipin came from the CCP4 monomer library (Winn et al, 2011). The model was validated using Mol-Probity (Chen et al, 2010). Visualisations of UCP1 structures were generated using PyMOL. Conservative modelling of both CTP and dTTP was conducted using both Coot and Isolde to fit each nucleotide into the UTP-bound cryo-EM map. Restraint files for CTP and dTTP came from the CCP4 monomer library (Winn et al, 2011). Structures of human OGC and DIC were generated using AlphaFold3 server (Abramson et al, 2024).

## Data availability

Coordinates and cryo-EM maps have been deposited in the Protein Data Base and Electron Microscopy Data Bank, with the following accession numbers: PDB: 9FZQ, EMD: 50894.

The source data of this paper are collected in the following database record: biostudies:S-SCDT-10_1038-S44318-025-00395-3.

## Peer review information

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

## Acknowledgements

We would like to acknowledge financial support from the UKRI Medical Research Council (MC_UU_00028/2) and UKRI Biotechnology and Biological Sciences Research Council (BB/Y002865/1). We would like to thank Cryo-EM Facility of the Biochemistry Department for access to the Vitrobot, and Dr Éilís Bragginton for assistance at the Cryo-EM facilities at the UK National Electron Bio-Imaging Centre (eBIC), proposal BI31589-47. Nanobody discovery was funded by the Instruct-ERIC (PID: 1270), part of the European Strategy Forum on Research Infrastructures (ESFRI), and the Research Foundation - Flanders (FWO).

## Author contributions

**Scott A Jones**: Conceptualization; Data curation; Formal analysis; Validation; Investigation; Visualization; Methodology; Writing—original draft; Writing—review and editing. **Alice P Sowton**: Data curation; Formal analysis; Validation; Investigation; Visualization; Methodology; Writing—original draft; Writing—review and editing. **Denis Lacabanne**: Data curation; Formal analysis; Validation; Investigation; Visualization; Methodology; Writing—review and editing. **Martin S King**: Data curation; Formal analysis; Validation; Investigation; Visualization; Methodology; Writing—review and editing. **Shane M Palmer**: Investigation; Methodology; Writing—review and editing. **Thomas Zögg**: Data curation; Formal analysis; Validation; Investigation; Methodology; Writing—review and editing. **Els Pardon**: Formal analysis; Validation; Investigation; Methodology; Writing—review and editing. **Jan Steyaert**: Conceptualization; Resources; Supervision; Funding acquisition; Investigation; Methodology; Project administration; Writing—review and editing. **Jonathan J Ruprecht**: Formal analysis; Validation; Investigation; Visualization; Methodology; Writing—review and editing. **Edmund R S Kunji**: Conceptualization; Resources; Formal analysis; Supervision; Funding acquisition; Validation; Methodology; Writing—original draft; Project administration; Writing—review and editing.

Source data underlying figure panels in this paper may have individual authorship assigned. Where available, figure panel/source data authorship is listed in the following database record: biostudies:S-SCDT-10_1038-S44318-025-00395-3.

## Disclosure and competing interests statement

The authors declare no competing interests.

