## [Peer Review File · The EMBO Journal]

Proton conductance by human uncoupling protein 1 is inhibited by purine and pyrimidine nucleotides

Scott Jones, Alice Sowton, Denis Lacabanne, Martin King, Shane Palmer, Thomas Zögg, Els Pardon, Jan Steyaert, Jonathan Ruprecht, and Edmund Kunji

Corresponding author(s): Edmund Kunji (ek@mrc-mbu.cam.ac.uk)

Review Timeline:

Submission Date:	16th Jul 24
Editorial Decision:	26th Aug 24
Appeal Received:	30th Oct 24
Editorial Decision:	29th Nov 24
Revision Received:	18th Dec 24
Editorial Decision:	24th Jan 25
Revision Received:	30th Jan 25
Accepted:	4th Feb 25

Editor: William Teale

Transaction Report:

Dear Edmund,

Thank you for submitting your study, "Proton conductance by human uncoupling protein 1 is inhibited by purine and pyrimidine nucleotides", to EMBO Journal; thank you also for your patience during the review process. Your work was reviewed by three referees, whose reports I have attached to the bottom of this email. I have read the reports and your manuscript very carefully and have discussed them with my editorial colleagues. As you will see, the reports are all unusually short. I take this as support for the technical quality of the work you present. However, as the situation stands, I don't have enough support from the referees to pursue this manuscript towards publication. My take is this: as long as the referees are unable clearly to ascertain the biological relevance of pyrimidine nucleotide inhibition from either a physiological or evolutionary perspective, the conceptual advance remains ill-defined.

I appreciate that, at its core, your work shows that pyrimidines are able to bind and inhibit UCP1. As all reviewers point out, your findings here are clearly novel and well supported by the data you present. However, I share referee 3's concerns that, without more progress into the extent to which pyrimidine-UCP1 binding integrates into the protein's cellular function (or its regulation by ATP), the work lacks context and, as a result, significance. If you can add these data, I will enthusiastically consider the manuscript again. For such resubmissions, we take novelty over the original manuscript into consideration and might involve additional referee(s). If you are considering a resubmission of the manuscript once you have gained further mechanistic insight, please contact me in advance.

Best wishes,

William

William Teale, PhD
Editor
The EMBO Journal
w.teale@embojournal.org

Referee #1:

This is a very solid study from an established and solid laboratory. While the conclusions are modest, they are well based and will add to the understanding of UCP1 structure and regulation. The primary finding is that UCP1 binds purines and pyrimidines while previously thought only to bind purines. Cryo-EM studies provide molecular details to the basis for the binding.

The authors also investigate the related mitochondrial oxoglutarate and dicarboxylate carriers and their ability to bind nucleotides. While these proteins are members of the SLC25 family of transporters and related to UCP1, they are shown not to bind nucleotides.

Overall, this is a solid study that provides new and important information on UCP1.

The paper is well written, but I would suggest one minor change in the text. In the final sentence in the Abstract should be rewritten. "Thus, UCP1 has evolved ... " Since UCP1 is a member of the SLC25 mitochondrial carriers, it isn't clear if the authors are claiming that this study demonstrates that UCP1 has evolved from the dicarboxylate carrier or if that was already known prior to this study. If this was already known, it might be better to say "Thus, while UCP1 has evolved..."

Referee #2:

The manuscript by Jones and colleagues presents surprising results that proton conductance through the human uncoupling protein UCP1 can be inhibited by both pyrimidine and purine nucleotides. It has been the accepted situation for 50 years that only purine nucleotides have this ability, results based primarily on studies in isolated brown fat mitochondria. While the results most probably will not lead to any change in our opinion that ATP (and ADP) are the physiological inhibitors, based on the cytosolic concentrations of nucleotides, the results are unexpected and extend our understanding of the amino acid residues involved in the nucleotide binding site in the central cavity of the protein. It could be relevant for the authors to comment why the studies with isolated mitochondria consistently indicate a lower ability of pyrimidine nucleotides to inhibit proton conductance than the results found here.

There have been several recent papers that have elucidated the structure of UCP1 that have confirmed the close relationship with other mitochondrial solute carriers which has allowed a description of the cycling process that permits proton conductance but without a full understanding of this pathway and its activation. Perhaps the new data can help extend our understanding of the mechanism of regulation through loss of the nucleotide inhibition.

Referee #3:

In this work, Jones SA et al determine that human uncoupling protein UCP1 can be inhibited not only by purine, but also by pyrimidine nucleotides. The authors determined thermostability, used isothermal titration calorimetry and cryo-EM to characterize the effect of nucleotide binding to highly purified, recombinant UCP1 to identify that all the tested nucleotides inhibit UCP1 activation by a shared mechanisms, which relies on binding to the same area of the protein. While the work is sound, well written and convincingly demonstrates pyrimidine nucleotide inhibition, the conceptual advance is limited. Moreover, it is clear, also from the authors discussion, that this novel finding might explain only a minority of the cases where UCP1 is inhibited, as ATP is the nucleotide with the highest cellular concentration. However, the mechanism by which this nucleotide-dependent inhibition is overcome during UCP1 activation is yet unknown. Identifying this mechanism would be important. Hence, in the absence of such significant finding, this work would be better suited for a more specialized journal.

** As a service to authors, EMBO Press provides authors with the possibility to transfer a manuscript that one journal cannot offer to publish to another EMBO publication or the open access journal Life Science Alliance launched in partnership between EMBO Press, Rockefeller University Press and Cold Spring Harbor Laboratory Press. The full manuscript and if applicable, reviewers' reports, are automatically sent to the receiving journal to allow for fast handling and a prompt decision on your manuscript. For more details of this service, and to transfer your manuscript please click on Link Not Available. **

Referee #1:

This is a very solid study from an established and solid laboratory. While the conclusions are modest, they are well based and will add to the understanding of UCP1 structure and regulation. The primary finding is that UCP1 binds purines and pyrimidines while previously thought only to bind purines. Cryo-EM studies provide molecular details to the basis for the binding.

The authors also investigate the related mitochondrial oxoglutarate and dicarboxylate carriers and their ability to bind nucleotides. While these proteins are members of the SLC25 family of transporters and related to UCP1, they are shown not to bind nucleotides.

Overall, this is a solid study that provides new and important information on UCP1.

We thank the reviewer for their supportive comments and for their acknowledgement of the important new insights on the regulation of UCP1 that our study provides.

The paper is well written, but I would suggest one minor change in the text. In the final sentence in the Abstract should be rewritten. "Thus, UCP1 has evolved ... " Since UCP1 is a member of the SLC25 mitochondrial carriers, it isn't clear if the authors are claiming that this study demonstrates that UCP1 has evolved from the dicarboxylate carrier or if that was already known prior to this study. If this was already known, it might be better to say "Thus, while UCP1 has evolved..."

UCP1 has been known to be phylogenetically related to both DIC and OGC and we have amended the sentence to clarify this.

Referee #2:

The manuscript by Jones and colleagues presents surprising results that proton conductance through the human uncoupling protein UCP1 can be inhibited by both pyrimidine and purine nucleotides. It has been the accepted situation for 50 years that only purine nucleotides have this ability, results based primarily on studies in isolated brown fat mitochondria.

We thank the reviewer for acknowledging the unexpected nature of our results, which have overturned a 50-year dogma in the UCP1 field.

While the results most probably will not lead to any change in our opinion that ATP (and ADP) are the physiological inhibitors, based on the cytosolic concentrations of nucleotides, the results are unexpected and extend our understanding of the amino acid residues involved in the nucleotide binding site in the central cavity of the protein. It could be relevant for the authors to comment why the studies with isolated mitochondria consistently indicate a lower ability of pyrimidine nucleotides to inhibit proton conductance than the results found here.

We agree the main nucleotides inhibiting UCP1 will likely be adenine nucleotides based on the known cytoplasmic concentrations of each nucleotide, as discussed and presented clearly in the manuscript. However, the concentrations of nucleotides in the mitochondrial intermembrane space are unknown and is unlikely to be the same as the cytoplasmic concentrations, as they are dependent on the flux caused by transport and enzymatic

activities, making the inhibition of UCP1 by pyrimidine nucleotides an important regulatory finding.

In light of the reviewer's remarks regarding the novelty of our finding, we have also added a further discussion point considering why these observations may have differed, which we believe are to do with the complexity of the mitochondrion with regards to nucleotide fluxes caused by enzymatic conversions and transport processes.

There have been several recent papers that have elucidated the structure of UCP1 that have confirmed the close relationship with other mitochondrial solute carriers which has allowed a description of the cycling process that permits proton conductance but without a full understanding of this pathway and its activation. Perhaps the new data can help extend our understanding of the mechanism of regulation through loss of the nucleotide inhibition.

We agree with the reviewer that this paper further expands on the recent papers structurally and biochemically characterising the nucleotide inhibition of UCP1. Our comparative analysis with OGC and DIC adds to our previous papers, highlighting D28 as a key residue in a potential proton translocation pathway. The comparison has also highlighted that UCP1 evolution did not select for base specificity of nucleotide binding, but for pH-dependency of binding of any nucleotide, which is mediated via the phosphate moieties. As we will argue below in response to reviewer 3, the pH-dependency is an integral part of the termination mechanism of the heat cycle to protect ATP synthesis, which is required for the survival of the cells.

Referee #3:

In this work, Jones SA et al determine that human uncoupling protein UCP1 can be inhibited not only by purine, but also by pyrimidine nucleotides. The authors determined thermostability, used isothermal titration calorimetry and cryo-EM to characterize the effect of nucleotide binding to highly purified, recombinant UCP1 to identify that all the tested nucleotides inhibit UCP1 activation by a shared mechanism, which relies on binding to the same area of the protein. While the work is sound, well written and convincingly demonstrates pyrimidine nucleotide inhibition, the conceptual advance is limited. Moreover, it is clear, also from the authors discussion, that this novel finding might explain only a minority of the cases where UCP1 is inhibited, as ATP is the nucleotide with the highest cellular concentration. However, the mechanism by which this nucleotide-dependent inhibition is overcome during UCP1 activation is yet unknown. Identifying this mechanism would be important. Hence, in the absence of such significant finding, this work would be better suited for a more specialized journal.

We thank the reviewer for acknowledging that the presented manuscript is "sound, well written and convincingly demonstrates pyrimidine nucleotide inhibition". Since this paper overthrows a dogma that has existed for half a century in this field (see reviewer 2), we reject the unfounded statement that the 'conceptual advance is limited'.

We have expanded our discussion to consider the physiological relevance of the regulation of nucleotide inhibition of UCP1 in the context of brown adipose tissue thermoregulation further by addressing two hypotheses. The first one has suggested that the increase in cytosolic calcium concentrations, which follow the adrenergic stimulation of BAT, terminates the inhibition by complexing the nucleotides. However, our new experimental data show that the binding of nucleotides to UCP1 is unaltered in the presence of all physiologically relevant

concentrations of Ca and Mg ions, rejecting this hypothesis. The reason is that all nucleotides are bound by six positively charged residues, which can effectively compete with divalent cations for binding of the phosphate moieties. The second hypothesis investigates the role of the pH-dependent mechanism in the inhibition of UCP1 activity, which we have shown in this paper to be true for all purine and pyrimidine nucleotides. We argue that the pH effect has nothing to do with the activation, but serves to limit the heat production, as uncoupling upon UCP1 activation leads to increased substrate oxidation, respiration and proton pumping, which in principle could lead to a runaway heat cycle. Instead, the increased proton pumping activity will acidify the intermembrane space, leading to an increased affinity of UCP1 for nucleotides and thus to termination of the proton leak to restore the balance between heat production and ATP synthesis. This mechanism explains why there has been no selective pressure on the nucleobase, as any nucleotide will do, because the pH-dependent mechanism is mediated via the phosphate moieties. Thus, this mechanism prevents a runaway heat cycle to allow ATP synthesis in order to support the other metabolic energy-requiring processes of BAT cells in a balanced manner.

Dear Edmund,

Thank you for your email. Sorry for the longer time taken than expected; I have been thinking about how to proceed for most of the week. I attach two re-review reports below. As you will see, reviewer #3 now appreciates the added context you have provided. However, reviewer #2 is concerned that the physiological basis of the discussion section has now been undermined.

I suggest we organise a Zoom call next week to see if a way through exists - could you find time on Tuesday or Wednesday morning?

Best wishes,

William

William Teale, PhD
Editor
The EMBO Journal
w.teale@embojournal.org

We realize that it is difficult to revise to a specific deadline. In the interest of protecting the conceptual advance provided by the work, we recommend a revision within 3 months (27th Feb 2025). Please discuss the revision progress ahead of this time with the editor if you require more time to complete the revisions. Use the link below to submit your revision:

Referee #2:

While I still find that the results of this study are of considerable interest for our understanding of the regulation of UCP1, I am unfortunately of the opinion that the revision has weakened the manuscript by including speculations that are without foundation and that are directly contrary to accepted Mitchellian theory. I cannot recommend publication if these issues are not corrected in an acceptable way, i.e. new revision is needed.

My criticism refers primarily to the later part of the new Discussion on pages 11 and 12 and to the new Figure 5. While it is clear that more protons will indeed be pumped out of the matrix when thermogenesis is high, this occurs because the proton motive force is decreased by activation of UCP1 and it is not, as is stated, maintained. This is the only way that the electron transport chain can increase its activity. It is unable to pump more protons against the high pmf. This means that the proton density in the intermembrane space will not increase because the protons will return rapidly to the matrix. Thus, the entire hypothesis advocated in the new discussion must be removed. - Additionally, (line 338), the authors have mixed up the formulation concerning the effects of a shift in pH; surely they mean the reverse (but a correction of this does not alter the main problem with the new discussion).

There is no reason to believe that the pmf would become very low during thermogenesis (as would be the case in the presence of an artificial uncoupler). Maximum respiratory capacity can be reached by merely decreasing the pmf from about 220 mV to about 170 mV. At this pmf, ATP synthesis can also occur.

The new Figure 5 is thus not acceptable as drawn, as it promotes an erroneous picture of Mitchellian bioenergetics. In addition, concerning the UCP1 part, in thermogenically inactive mitochondria in situ, and based on known nucleotide concentrations, it must be assumed that UCP1 has a nucleotide bound and is thus inactive. Following initiation of thermogenesis, it is most likely that fatty acids released from lipolysis activate UCP1. However, this activation does not lead to release of the nucleotide from UCP1 (Nicholls). Admittedly, the activation process is unclear but cannot be drawn as indicated.

Furthermore, the response to my question about studies in isolated mitochondria led to a handwaving response about nucleotide concentrations in the intermembrane space "making the inhibition of UCP1 by pyrimidine nucleotides an important regulatory finding", which is not a motivated response.

Thus, without omission of these new parts of the paper (or very major reformulations that do not seem possible), the paper is not acceptable.

Referee #3:

In this revised manuscript, the authors have now included further data and discussions on the mechanism of nucleotide-based regulation of UCP1 in the context of BAT. These data and the proposed model for how nucleotide-based inhibition limits UCP1 activation allow to put the presented work into a larger context, rendering the work more interesting to a larger audience. To further elaborate their model, the authors could discuss one aspect, namely how varying concentrations of FFAs together with fluctuating pH values in the IMS could achieve fine-tuned regulation.

Minor point:

Line 326: election should be electron

Referee #2:

While I still find that the results of this study are of considerable interest for our understanding of the regulation of UCP1, I am unfortunately of the opinion that the revision has weakened the manuscript by including speculations that are without foundation and that are directly contrary to accepted Mitchellian theory. I cannot recommend publication if these issues are not corrected in an acceptable way, i.e. new revision is needed.

My criticism refers primarily to the later part of the new Discussion on pages 11 and 12 and to the new Figure 5. While it is clear that more protons will indeed be pumped out of the matrix when thermogenesis is high, this occurs because the proton motive force is decreased by activation of UCP1 and it is not, as is stated, maintained. This is the only way that the electron transport chain can increase its activity. It is unable to pump more protons against the high pmf. This means that the proton density in the intermembrane space will not increase because the protons will return rapidly to the matrix. Thus, the entire hypothesis advocated in the new discussion must be removed. - Additionally, (line 338), the authors have mixed up the formulation concerning the effects of a shift in pH; surely they mean the reverse (but a correction of this does not alter the main problem with the new discussion).

There is no reason to believe that the pmf would become very low during thermogenesis (as would be the case in the presence of an artificial uncoupler). Maximum respiratory capacity can be reached by merely decreasing the pmf from about 220 mV to about 170 mV. At this pmf, ATP synthesis can also occur.

We agree with the reviewer that the pmf in thermogenesis, following the activation of UCP1 by fatty acids, is not fully dissipated, but maintained to a lower level. This is an essential requirement, as heat can only be generated by energy conversion from the pmf. It is consistent with an increase in oxygen consumption, meaning that the proton pumping rate of the respiratory chain is increased too, because it has to compensate for the proton leak. We agree with the reviewer that the local pH in the intermembrane space under these conditions is unknown, as it depends on the relative rates of proton pumping and leak, and also on other processes, such as the activity of ATP synthase and transporters. In the inhibited state, the flux of protons through UCP1 is nil, but upon fatty acid activation the flux of protons through UCP1 increases significantly, as part of the uncoupling mechanism. With increased proton flux, the probability of the proton-mediated termination by nucleotide binding does also increase, providing a negative feedback loop. It is not an all-on or all-off mechanism, but more like a dimmer switch, utilising the population of UCP1 by tempering uncoupling through inhibition, followed by reactivation by fatty acids in a stochastic process. This could explain the fundamental difference with chemical (artificial) uncoupling, which has no tempering mechanism.

The new Figure 5 is thus not acceptable as drawn, as it promotes an erroneous picture of Mitchellian bioenergetics. In addition, concerning the UCP1 part, in thermogenically inactive mitochondria in situ, and based on known nucleotide concentrations, it must be assumed that UCP1 has a nucleotide bound and is thus inactive. Following initiation of thermogenesis, it is most likely that fatty acids released from lipolysis activate UCP1. However, this activation does not lead to

release of the nucleotide from UCP1 (Nicholls). Admittedly, the activation process is unclear but cannot be drawn as indicated.

Well, Michellian bioenergetics is violated in uncoupling, as one of the postulates, i.e. the assumption of an impermeable membrane, no longer holds. As we said in the original version, the activation mechanism of UCP1 is unknown and thus the interplay between fatty acid activation and nucleotide inhibition of UCP1 is currently unclear. We have removed Fig 5, because of this reason, but also because it is nearly impossible to convey a probability-based mechanism of inhibition, as described above, in simple pictorial terms.

Furthermore, the response to my question about studies in isolated mitochondria led to a handwaving response about nucleotide concentrations in the intermembrane space "making the inhibition of UCP1 by pyrimidine nucleotides an important regulatory finding", which is not a motivated response.

It is unclear to us why inhibition by pyrimidine nucleotides was not observed in mitochondrial studies of proton leak, but we made some suggestions in the manuscript.

Thus, without omission of these new parts of the paper (or very major reformulations that do not seem possible), the paper is not acceptable.

We have removed Figure 5 and re-written our hypothesis with regards to the pH dependent mechanism of inhibition, following the comments of the reviewer for which we are grateful.

Referee #3:

In this revised manuscript, the authors have now included further data and discussions on the mechanism of nucleotide-based regulation of UCP1 in the context of BAT. These data and the proposed model for how nucleotide-based inhibition limits UCP1 activation allow to put the presented work into a larger context, rendering the work more interesting to a larger audience. To further elaborate their model, the authors could discuss one aspect, namely how varying concentrations of FFAs together with fluctuating pH values in the IMS could achieve fine-tuned regulation.

We thank the reviewer for their comments. The problem is that we do not fundamentally understand the interplay between fatty acid activation and nucleotide inhibition. Following advice from reviewer 2 we have removed the pH comments as we cannot be sure what the pH is under these conditions (subject of many processes), but we can be sure that the flux of protons increases substantially upon activation, which might trigger the proton-mediated nucleotide inhibition as part of a tempering mechanism. We also kept the part showing a lack of calcium or magnesium ion effect on inhibitor binding.

Minor point:

Line 326: election should be electron

This error has been corrected.

Dear Edmund,

We have now received a re-review report from referee #2, which I have included below. As you will see, there is little movement on the point of acidification of the mitochondrial inter-membrane space (IMS). I suggest that (as you clearly point out) as there are no unambiguous IMS pH data available in this context, you follow the advice of referee #2 and further temper this section of the discussion. All reviews and your responses to them will be freely available in files associated with the article, preserving your discussion for the record. This will no-doubt be a valuable addition to the debate.

Before I can accept the manuscript, there remain some editorial points which need to be addressed. In this regard would you please:

- move funding information in the Comments box to the 'More Funders' list as our production team could not extract this information,
- include up to five key words,
- include a data-availability section with links to publicly-available data sets,
- include a "Disclosure and competing interests statement",
- remove the AC/CrediT section from the text,
- include callouts for Table S1 (Appendix Table S1) and Appendix Figures S1 and S5,
- upload the appendix file in PDF format; title page should contain "Appendix for Proton conductance by human uncoupling protein 1 is inhibited by purine and pyrimidine nucleotides" and a table of contents with the page numbers for the listed items; use the nomenclature 'Appendix Figure Sx' and 'Appendix Table Sx' throughout the manuscript and Appendix PDF,
- upload Source data files in a scheme one figure/folder as .zip files. E.g. all the Source data files for figure 1 need to be saved in a single folder and this needs to be zipped and then uploaded as "SD figure 1.zip" file,
- provide the completed data availability statement,
- provide exact p values in the legends of figures 1C and 2F, H
- correct the section as follows: Title page - Abstract & Keywords - Introduction - Results - Discussion - Methods - Data Availability - Acknowledgements - Disclosure and Competing Interests Statement - References - Figure Legends - Table(s) - Expanded View Figure Legends.

We include a synopsis of the paper (see <http://emboj.embopress.org/>). Please also therefore provide me with a general summary image, a two sentence statement and 3-5 bullet points that capture the key findings of the paper.

I am looking forward to receiving your revised manuscript.

EMBO Press is an editorially independent publishing platform for the development of EMBO scientific publications.

Best wishes,

William

William Teale, PhD
Editor
The EMBO Journal
w.teale@embojournal.org

- a point-by-point response to the referees' comments, with a detailed description of the changes made (as a word file).
- a word file of the manuscript text.
- individual production quality figure files (one file per figure)
- a complete author checklist, which you can download from our author guidelines (<https://www.embopress.org/page/journal/14602075/authorguide>).

- Expanded View files (replacing Supplementary Information)

- a Reagents and Tools Table as part of the Methods section, which can be downloaded from our author guidelines

(<https://www.embopress.org/page/journal/14602075/authorguide#structuredmethods>)

We realize that it is difficult to revise to a specific deadline. In the interest of protecting the conceptual advance provided by the work, we recommend a revision within 3 months (24th Apr 2025). Please discuss the revision progress ahead of this time with the editor if you require more time to complete the revisions. Use the link below to submit your revision:

Referee #2:

In both the response and in the Discussion (l.339 - 340), the authors insist that an increased H⁺ flux should lead to increased "termination". This is not understandable and this means that the "key role" ascribed (l.332) to pH is not supported by data or theory. It is my advice that this section (l. 331-345) is omitted.

All editorial and formatting issues were resolved by the authors.

Dear Edmund,

I am pleased to inform you that your manuscript has been accepted for publication in the EMBO Journal.

Congratulations to you and your team on a really insightful study!

Best wishes,

William

William Teale, PhD
Editor
The EMBO Journal
w.teale@embojournal.org
